# Peer review of "Abundance, Distribution, and Habitat Preference of Syngnathid Species in Sabaudia Lake (Tyrrhenian Sea)"

_diversity, doi:10.3390/d15090972_

Round 1
Reviewer 1 Report
This paper provides a first look at the diversity, distribution and abundances of syngnathids within Sabaudia lake and the information presented here provides an important baseline. I do, however, think that a bit more could have been done with the data – e.g. placing the research in a wider context, presenting densities (for easy comparisons with other studies) etc. I also think more detail can be provided with regards to the survey design and handling of the unbalanced sampling effort across sites. Even though habitat is characterised, no assessment was conducted to determine what habitat features drive syngnanthid presence – so habitat association/preference conclusions cannot really be made (or at least, this was not clear to me). Detailed comments are provided below.
Abstract - It might be useful to give some results in the abstract – maybe average densities.
Line 40 – I don’t think the fact that syngnathids are assessed via the IUCN Red List is because of the threats they face.
Line 41 – This sentence is a bit confusing. I understand that information regarding abundances, distributions etc. are needed for this group, and that they act as flagship species. Perhaps reread the sentence to make sure it is clear.
Line 54 – I am always careful in using the term preference, when talking about a non-experimental assessment. Preference needs to be tested via an experimental approach. I would rather use the word association for this type of investigation.
Line 60 – The introduction is very short and very focused only on syngnathids in the Mediterranean. Which is not a problem per se – but I would think it would be valuable to provide some wider context and value – thinking about expanding on the flagship status of these fishes, the threats on coastal environmental, the need for basic ecological information for species conservation. All aspects were touched on – but I think more can be made of it.
Figure 1 – I would add letters to the figure inserts and provide a bit more details/descriptions in the legend.
Line 81 – This information needs a reference.
Line 90 – It would be helpful to give the area of the lake.
Line 93 – When was the preliminary survey conducted? Did it cover the entire lake?
Line 97 – So, at every site, two 50m transects were surveyed? What was the area of each transect? And where abundances/densities averaged across the transects?
Line 103 – I would add some more details of the habitat assessment approach, and not just the reference.
Figure 2 – I think this information needs additional description in the text. Introduce the different habitats, how they were classified/identified, how the different areas for each habitat were determined etc. I am also unsure how the number of stations within each habitat type were determined (it does not seem balances across, with more effort in some habitats compared to others).
Line 111 – Here it seems the stations were grouped in a different manner? It is not clear how analyses were done, how differences in efforts across habitats were handled or what factors were compared in the end. I think this section can do with additional information.
Line 131 – I don’t think microhabitat is the correct term to use when talking about the different habitats.
Table 1 – For the different habitats, was this a general assessment of the entire station (and if so, what is the size of the area); or does it only refer to what was underneath the transect area? I would also write the scientific names in full for the first time at least.
Line 143 – Was Dissolved Oxygen measured? And if not, how was the anoxic nature of the habitat determined?
Figure 4 – I think mean sizes with a SE should be shown in the figure. The figure at the moment does not show size distribution. And if the sizes of fish were not compared across sites – why have the different sites shown in the figure?
Figure 5 – I am not sure what is being shown in this figure? Is it the mean density per site? Mean abundance? The axis labels are also not fully in the figure. I am also not sure about the reference to sampling stations.
Figure 6 – I think the approach taken here needs to be described in a bit more details in the methods – and how the differences in effort across sites etc. were handled.
Line 185 – Why not present mean densities of syngnathids across the different habitats? This can then be compared to other studies and provide more standardised data.
Line 194 – How many species out of the total number are listed as DD?
Table 2 – I am not too sure of the relevance of this table?
Line 210 – Usually, a discussion should start with a paragraph like this.
Line 211 – I am not sure I understand this sentence?
Line 217 – A reference is needed for this statement.
Line 224 – This is repetitive of what was stated in the introduction.
Line 243 – Are habitats with a lot of holdfasts not also more complex?
Line 247 – I am not sure of the relevance of this paragraph. Water quality was not assessed in this study.
Line 278 – This sentence is a bit contradictory.
The quality of English is fair and will benefit from some editing.
Author Response
Thanks for the critical review of the manuscript. We realized that there was confusion in the use of some terms which generated confusion in the interpretation of the text. We have taken steps to make the manuscript more readable.

Reviewer 2 Report
The research work presented is of great importance for the group (Syngnathidae), as it identifies and quantifies a moment in the life of near-threatened (NT) and data-deficient (DD) species. The work is well conducted and with a clear objective of investigating the abundance, distribution and preferred habitats of the species.
As it was a short-term research, there is no room for major discussions, since there is no series of data over time, which does not diminish the importance of the work carried out.
Here is a contribution to the article:
Figure 4. It would be interesting to identify the “x” and “y” axes. Am I understanding that in the “x” are the seasons and in the “y”? According to the caption it would be the size of the fish, but it seems that it is not...
Figure 5. Are the captions of figures 4 and 5 interchanged?
I suggest that the Discussion should start at line 210, the previous text could be removed for introduction and Table 2 could be supplementary material.
On lines 265-66 it says that the low complexity/muddy habitats are preferred by H. hippocampus, however in Figure 3 this habitat is showing H. gutullatus, and the more complex one shows H. hippocampus. I believe that both species share both types of habitats, but I suggest adapting the text or photo.
The work done was very good and I wish the authors good luck with future long-term studies that can be carried out with different methodologies for seahorses and pipefish.
Author Response

(The authors gave the same response as above.)
